# The EphA1 and EphA2 Signaling Modulates the Epithelial Permeability in Human Sinonasal Epithelial Cells and the Rhinovirus Infection Induces Epithelial Barrier Dysfunction via EphA2 Receptor Signaling

**DOI:** 10.3390/ijms24043629

**Published:** 2023-02-11

**Authors:** Jae Min Shin, Moon Soo Han, Jae Hyung Park, Seung Hyeok Lee, Tae Hoon Kim, Sang Hag Lee

**Affiliations:** Department of Otorhinolaryngology-Head & Neck Surgery, College of Medicine, Korea University, Seoul 02841, Republic of Korea

**Keywords:** chronic rhinosinusitis, ephrinA1, ephrinA2 receptor, rhinovirus, epithelial barrier

## Abstract

Deficiencies in epithelial barrier integrity are involved in the pathogenesis of chronic rhinosinusitis (CRS). This study aimed to investigate the role of ephrinA1/ephA2 signaling on sinonasal epithelial permeability and rhinovirus-induced epithelial permeability. This role in the process of epithelial permeability was evaluated by stimulating ephA2 with ephrinA1 and inactivating ephA2 with ephA2 siRNA or inhibitor in cells exposed to rhinovirus infection. EphrinA1 treatment increased epithelial permeability, which was associated with decreased expression of ZO-1, ZO-2, and occludin. These effects of ephrinA1 were attenuated by blocking the action of ephA2 with ephA2 siRNA or inhibitor. Furthermore, rhinovirus infection upregulated the expression levels of ephrinA1 and ephA2, increasing epithelial permeability, which was suppressed in ephA2-deficient cells. These results suggest a novel role of ephrinA1/ephA2 signaling in epithelial barrier integrity in the sinonasal epithelium, suggesting their participation in rhinovirus-induced epithelial dysfunction.

## 1. Introduction

Chronic rhinosinusitis (CRS) is a chronic heterogeneous inflammatory disorder in sinonasal mucosa, with pathophysiological causes that remain unclear. Substantial progress has been made toward understanding the inflammatory mechanisms associated with CRS. Recent studies have suggested that dysregulation in the innate immune response of the sinonasal epithelium, including disrupted epithelial barrier integrity, is involved in the initial inflammatory response and subsequently enhances the activation of inflammatory cells, resulting in a chronic inflammatory response [1,2,3,4,5].

The sinonasal epithelium acts as a protective barrier against pathogens or particles inhaled into the nasal cavities. To manage this task, the nasal epithelial cells are comprised of tight junctional complexes formed between neighboring cells, inhibiting the paracellular movement of solute and water [2,6]. Actually, the sinonasal epithelium plays critical roles in defense against viral respiratory tract infection. Rhinovirus (RV) is responsible for the majority of upper airway infections and can disrupt the junctional proteins during the early phase of infection, leading to epithelial barrier disintegrity [7,8,9,10,11]. Dysfunctional immune response of epithelial cells modulated by a viral infection can facilitate subsequent bacterial infection, enhancing the perpetuation of CRS symptoms [12]. A better understanding of biological factors attenuating rhinoviral infection may lead to the suppression of disease progression.

An interesting group in this regard comprises the erythropoietin-producing human hepatocellular carcinoma (eph) receptor tyrosine kinases and their ligands ephrins [13]. Based on ligand binding preferences, the ephrin receptor family is divided into two broad classes, ephA and ephB receptors, which are involved in various biological functions, such as cell assembly, migration, and adhesion [13]. Specifically, ephrinA1 and ephA2 are recognized as key regulators of inflammation [13]. The Eph receptors, components of receptor tyrosine kinases, and ephrins participate in the immune responses against pathogens, and ephA2 serves as the receptor for the invasion of microorganisms, including helicobacter pylori and Epstein-Barr virus, participating in host–pathogen interaction [13,14,15,16,17,18]. EphA2 deficiency suppressed pulmonary inflammation by attenuating endothelial leakage, epithelial permeability, and other inflammatory responses [19,20,21]. The ephrinA1/ephA2 interaction regulates the expression of the adherens junction, modulating the intestinal barrier integrity [22]. Our previous study showed that the expression of the ephrinA1/ephA2 receptor increases in chronic rhinosinusitis, and ephrinA1/epA2 signaling affects rhinovirus-induced innate immunity in human sinonasal epithelial cells [23]. However, whether they are involved in the maintenance of sinonasal epithelial barrier integrity has not been investigated.

Based on these previous data, we seek to evaluate whether the ephrinA1/ephA2 pathways participate in sinonasal epithelial barrier integrity and in rhinovirus-induced epithelial permeability.

## 2. Results

We evaluated if ephrinA1/ephA2 signaling participates in epithelial permeability through treatment with ephrinA1. The epithelial permeability and electrical resistance induced by ephrinA1 treatment were regulated in a dose-dependent manner (Figure 1A,B). Epithelial permeability was markedly increased by ephrinA1 treatment, which was accompanied by decreased TEER compared with the untreated control (Figure 1A,B). EphrinA1-induced epithelial permeability and transepithelial electrical resistance were attenuated in cells with the blockage of ephA2 action, which was accomplished by ephA2 siRNA or ephA2 inhibitor (Figure 1C,D).

To determine whether epithelial barrier disruption is associated with changes in intercellular junctions, the expression and distribution of ZO-1, ZO-2, and occludin were evaluated in cultured epithelial cells stimulated with ephrinA1 using western blotting and confocal microscopy. Western blot analysis demonstrated that the level of ZO-1, ZO-2, and occludin is lower in ephrinA1-treated cells than those of the control (Figure 2A). The effect of ephrinA1 was also suppressed by attenuating the action of the ephA2 receptor using the transfection of ephA2 siRNA or the pretreatment with an ephA2 inhibitor (Figure 2A). In parallel with the results of western blotting, confocal microscopy demonstrated the discontinuous localization of ZO-1, ZO-2, and occludin in ephrinA1-treated epithelial cells, whereas their distribution pattern was intact in cells that were transfected with ephA2 siRNA or pretreated with ephA2 inhibitor (Figure 2B).

Next, the expression of ephrinA1, ephrinA2, and the phosphorylated form of ephA2 was assessed in RV-infected epithelial cells. The data showed that the levels of ephrinA1 and ephA2 expression increased in RV-infected cells (Figure 3A). Furthermore, phosphorylated ephA2 level was also up-regulated in RV-infected cells, but not in cultured cells. where the ephA2 action was blocked by transfecting with ephrinA2 siRNA or pretreatment with ephA2 inhibitor (Figure 3B). These results suggest the possibility that ephrinA1 and ephA2 signaling can participate in RV-induced inflammatory response.

As described previously [11,24], in the present study, a disrupted epithelial barrier associated with decreased TEER was observed in cells infected with RV. These findings were also attenuated by suppressing the action of ephA2 (Figure 4A,B). Furthermore, we investigated the expression and distribution pattern of junctional complex proteins in RV-infected cells to determine the underlying mechanism of epithelial barrier damage induced by RV exposure. The levels of ZO-1, ZO-2, and occludin were decreased in RV-infected cells in comparison to those of the control (Figure 4C). This down-regulation was also reversed by blocking the action of ephA2, which was confirmed by confocal microscopic findings (Figure 4D). These data suggest that the disruption of the epithelial barrier induced by RV infection may be mediated through the ephA2 receptor.

## 3. Discussion

Recent studies demonstrated that diminished epithelial barrier in patients with CRS may compromise the interaction between the host and external immune stimuli [25]. Nevertheless, the regulation of epithelial barrier function and junctional protein expression has not been extensively studied in patients with CRS. In the current study, the role of ephrinA1/ephA2 signaling in epithelial permeability was assessed in the sinonasal mucosa. The stimulation of sinonasal epithelial ephA2 receptors with ephrinA1 induced increased epithelial permeability accompanied by decreased TEER. Furthermore, in epithelial cells infected with rhinovirus, the levels of ephrinA1, ephA2 receptor, as well as the phosphorylated form of ephA2 receptor were up-regulated and were associated with increased epithelial permeability. These effects were suppressed by attenuating the action of ephA2 with transfection with ephA2 siRNA or pretreatment with an ephA2 inhibitor. Taken with these results, ephrinA1/ephA2 pathways could participate in the maintenance of the epithelial barrier integrity and contribute to modulating rhinovirus-induced epithelial permeability.

Our results confirm previously reported data that RV infection downregulates the expression of junctional proteins in sinonasal epithelial cells, resulting in increased transepithelial permeability associated with reduced transepithelial resistance [11,26]. Consistent with these results, ephrinA1 stimulation of the sinonasal epithelial ephA2 receptor led to increases in epithelial permeability associated with decreased TEER, which mimicked the actions of RV. Our current data support previous studies that ephrin A1/ephA2 pathways participate in modulating vascular permeability in the lung and epithelial permeability in the colon [19,20,21,22]. Although the regulation of epithelial barrier integrity is not completely understood, transepithelial permeability is believed to be largely controlled by the state of intercellular junctions. Tight junctions are the main structures in determining sinonasal epithelial barrier integrity [24,25,27]. The expression of ZO-1 and occludin reduced in the epithelial cell layer has been associated with increased epithelial permeability of sinonasal mucosa [28,29]. Therefore, we evaluated whether epithelial barrier disruption induced by eprhinA1 or RV infection could be the result of ephA2 receptor-mediated rearrangement of ZO-1, ZO-2, and occludin. Here, confocal microscopic findings demonstrated that ephrinA1 treatment or RV infection enhanced patchy loss of ZO-1, ZO-2, and occludin immunoreactivity in epithelial cells, which was verified by western blotting. Furthermore, increased epithelial permeability enhanced in RV-infected and ephrinA1-treated epithelial cells was suppressed by blocking ephrinA2 activity with transfection of ephA2 siRNA or ephA2 inhibitor treatment, leading to increased TEER and well-preserved distribution of ZO-1, ZO-2, and occludin. Collectively, these results suggest that the ephA2 receptor could mediate epithelial hyperpermeability of sinonasal mucosa induced by RV infection, acting as a potential target. These findings were consistent with the results showing that blocking the activation of the ephA2 receptor by ephA2 mAb suppressed the endothelial permeability [30]. Furthermore, lipopolysaccharide (LPS) treatment enhanced the reduced expression of E-cadherin, which was restored in cells pretreated with ephA2 mAb [30]. LPS disrupted intestinal epithelial barrier integrity, with dismissed expression of occludin and claudin-1 and decreased TEER. However, the treatment with ephA2 mAb suppressed LPS-induced epithelial permeability [22]. EphrinA1 treatment was associated with disruption of tight and adherence junctions in pulmonary vessels [31]. Tobacco smoke exposure enhanced the upregulation of ephA2 and ephrinA1 levels in bronchial epithelial cells, decreased the expression of E-cadherin, and increased epithelial permeability. But the blockage of the ephA2 receptor with ephA2 siRNA attenuated the tobacco smoke–induced hyperpermeability [21]. Further studies are required to clarify these issues.

## 4. Materials and Methods

### 4.1. Patients and Sampling 

Sinonasal mucosa was acquired from the ethmoid sinus of patients with blowout fractures and the uncinate process of patients with septal deviation during surgery. The mucosal samples obtained during surgery were used for epithelial cell culture. The review board and the ethical committees of our hospital approved this study protocol. Written informed consent was provided by all participants. Subjects with history of a previous sinus surgery or viral URI, diagnosed with allergic rhinitis and asthma, or using antihistamines, steroids, or antibiotics 3 months before surgical treatment were excluded. Viral upper airway infection was evaluated according to a symptom scoring system developed by Jackson et al. [32].

### 4.2. Epithelial Cell Culture

Sinus mucosal samples obtained during surgery were incubated with 0.5% dispase (GenDEPOT, Katy, TX, USA) for 24 h. Thereafter, epithelial cells were isolated from mucosa by mechanical detachment using a cell scraper and cultured in culture plates containing Bronchial epithelial cell medium (BEpiCM; ScienCell, Carlsbad, CA, USA). After the culture surface was fully occupied by epithelial cells, cells harvested with trypsin were cultured under an air–liquid interface (ALI) in the SPL Insert system (SPL, Gyeonggi-do, Republic of Korea). ALI culture was used for all experiments.

### 4.3. RV Propagation and Infection

Human RV 16 (ATCC VR-283PQ) was generated by infecting H1HeLa cells (ATCC, Manassas, VA, USA) in flasks containing EMEM (ATCC) at 33 °C/5% CO_2_. H1HeLa cells infected with RV 16 were examined for cytopathic effect, and viral titer was evaluated by tissue culture infectivity dose (TCID_50_) assay. 

### 4.4. Effect of RV Infection on Expression of EphrinA1 and EphA2 Receptor

After inoculation with human RV 16 at an MOI of 0.5, cultured epithelial cells were incubated for 4 h in an incubator with 5% CO_2_ at 33 °C. After aspiration of non-attached viruses, cultured cells were incubated for 48 h at 33 °C. Thereafter, the levels of ephrinA1, ephA2, and the phosphorylated form of ephA2 were assessed through western blot.

### 4.5. Effect of EphrinA1/EphA2 Pathway on the Maintenance of the Epithelial Barrier Function

Epithelial barrier function was assessed by evaluating both transepithelial permeability and transepithelial electrical resistance (TEER). To block the action of the ephA2 receptor, ephA2 siRNAs were transfected into cultured epithelial cells or an ephA2 inhibitor (1 μM, ALW-II-41-27, APExBIO, Houston, TX, USA) was added to the culture media for 48 h. Thereafter, cultured cells were treated with ephrinA1 at a dose of 1, 5, 10, and 15 ng/mL or infected with RV for 48 h. Epithelial permeability and TEER were assessed using FITC-dextran (Sigma-Aldrich, St. Louis, MO, USA) and the ERS-2 volt-ohm meter (Millipore, Billerica, MA, USA), respectively. Permeability was analyzed by the application of fluorescein isothiocyanate (FITC)-dextran 4 kDa (Sigma-Aldrich) to the apical surface of cultured cells. Two hours after FITC-dextran was added apically at a concentration of 10 mg/mL, the passage of dextran into the basolateral fluids was analyzed using a Fluoroskan Ascent FL2.5 reader (Thermo Fischer, Loughborough, UK). The concentration of FITC-dextran was expressed as the ratio of the concentration in cells treated with rhinovirus relative to the value for untreated control cells. The values of epithelial permeability and TEER were expressed as a percentage of the control value.

The sinonasal epithelium acts as a protective barrier against pathogens or particles inhaled into the nasal cavities. To manage this task, the nasal epithelial cells comprise tight junctional complexes formed between neighboring cells, including Zonula occludens (ZO)-1, ZO-2, and occludin [2,6]. To evaluate the effect of the ephrinA1/ephA2 pathway on the expression of tight junctional proteins, the expression and distribution of ZO-1, ZO-2, and occludin (Invitrogen, Waltham, CA, USA) were evaluated by western blotting and confocal microscopy in cultured cells after measurement.

### 4.6. Small Interfering RNA (siRNA) Transfection Experiments

We transfected cultured cells with small interfering RNS (siRNA) directed towards ephA2 siRNA using Lipofectamine 2000 (Invitrogen, Waltham, CA, USA). Human ephA2 siRNA sequences were constructed as follows: AGUAGAGGUUGAAAGUCU and GAGACUUUCAACCUCUAC. The blocking effect of ephA2 siRNA was tested by RT-qPCR and western blotting. The MTT assay kit (Abcam, Cambridge, UK) was used to assess cell viability.

### 4.7. Western Blot

For western blot analysis, the proteins were extracted from frozen mucosal samples and cultured epithelial cells with radio-immunoprecipitation assay (RIPA) buffer (GenDEPOT). Denatured proteins were fractionated on SDS-PAGE and transferred to PVDF membranes (BioRad, Bedford, MA, USA). Thereafter, The membrane was rinsed and probed with primary antibodies in the refrigerator; the primary antibodies employed for western blots were anti-ephrinA1 antibody (1:250) obtained from Santa Cruz Biotechnology, anti-ephrinA2 antibody (1: 250, Abcam, Cambridge, UK), anti-phosphorylated ephA2 (1:1000, Cell Signaling Technologies, Danvers, MA, USA), anti-ZO-1 antibody (1:1000), anti-ZO-2 antibody (1:1000), anti-occludin antibody (1:1000), which were obtained from Invitrogen.

### 4.8. Confocal Microscopy

Confocal microscopy was performed to evaluate the localization of ZO-1, ZO-2, and occludin in cultured epithelial cells with anti-ZO-1, anti-ZO-2, and anti-occludin antibodies (Invitrogen).

### 4.9. Statistical Analyses

Statistical analyses were carried out using SPSS for Windows (version 16.0.0; SPSS, Chicago, IL, USA). One-way analysis of variance was conducted for the comparison of the data obtained by western blotting. Multiple comparisons were conducted by the Kruskal–Wallis test with Bonferroni *post-hoc* test. Data are expressed as the means ± standard error (SEM). Significant differences were set at <0.05.

## 5. Conclusions

EphrinA1 treatment increased epithelial permeability via the ephA2 receptor. Furthermore, rhinovirus infection also increased epithelial permeability associated with up-regulated levels of ephrinA1 and ephA2 receptors. These effects were suppressed by attenuating the action of ephA2. Therefore, the ephrinA1/ephA2 pathways could participate in the maintenance of epithelial barrier integrity in sinonasal mucosa, providing new therapeutic targets for rhinovirus-induced epithelial permeability. However, some limitations exist in this study. The effect of ephrinA1 and ephA2 receptor pathways on the epithelial permeability needs to be confirmed in in vivo experiments, including animal studies.

## Figures and Tables

**Figure 1 ijms-24-03629-f001:**
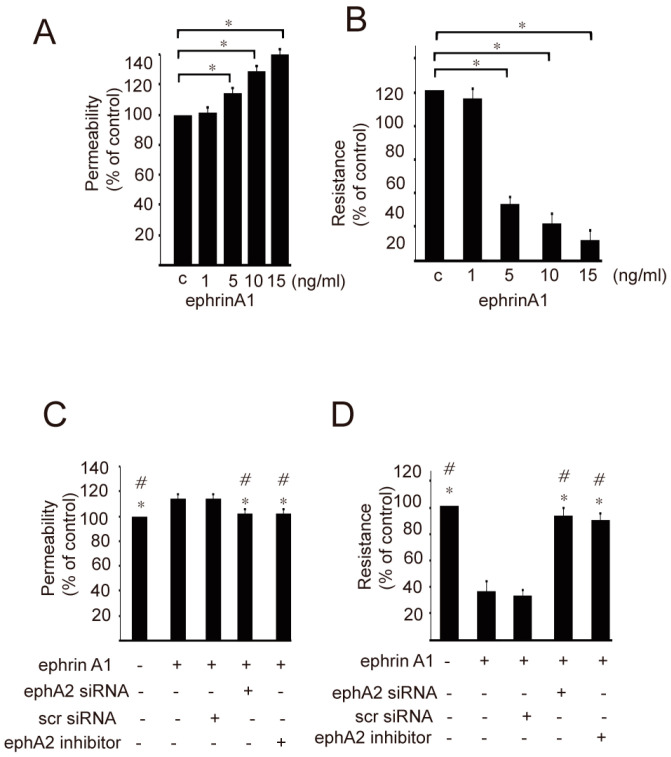
The epithelial permeability (**A**) and transepithelial electrical resistance (TEER, (**B)**) induced by ephrinA1 treatment at doses of 1, 5, 10, and 15 ng/mL were regulated in dose-dependent manner. The increased epithelial permeability (**C**) associated with decreased TEER (**D**) was observed in cultured epithelial cells treated with ephrinA1 (10 ng/mL) alone and in cells pretreated with scr siRNA followed by ephrinA1. However, the increased epithelial permeability associated with decreased TEER in cells treated with ephrinA1 was attenuated in cells pretreated with ephA2 siRNA or ephA2 inhibitor followed by ephrinA1 treatment. The values of epithelial permeability and TEER were expressed as a percentage of the control value. Data from seven independent experiments are presented as means ± standard error of the mean (SEM). * in (**A**,**B**) indicates *p* < 0.05. * and # in (**C**,**D**) indicate statistical significance *p* < 0.05 versus ephrinA1 or ephrinA1 + scr siRNA (scrambled siRNA).

**Figure 2 ijms-24-03629-f002:**
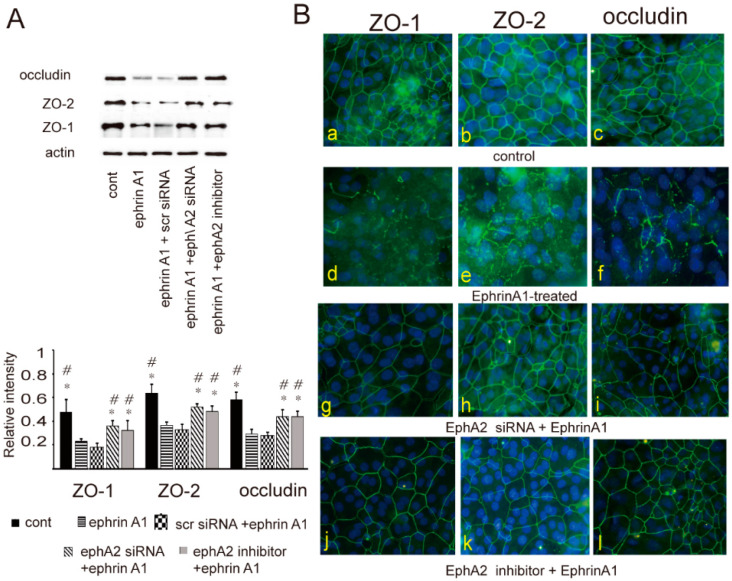
(**A**) Western blots show that the expression levels of ZO-1, ZO-2, and occludin were decreased in cultured cells treated with ephrinA1 (10 ng/mL, compared with those of the untreated control, which was reversed by blocking ephA2 with ephA2 siRNA or ephA2 inhibitor. Data from seven independent experiments are presented as means ± standard error of the mean (SEM). * and # in (**A**) indicate statistical significance *p* < 0.05 versus ephrinA1 or ephrinA1 + scr siRNA (scrambled siRNA). (**B**) Confocal microscopic findings; the distribution pattern of ZO-1 (**a**,**d**,**g**,**j**), ZO-2 (**b**,**e**,**h**,**k**), and occludin (**c**,**f**,**i**,**l**) in untreated epithelial cells (control, **a**–**c**), ephrinA1-treated cells (**d**–**f**), in cells transfected with ephA2 siRNA followed by ephrinA1 (**g**–**i**), and in cells treated with ephA2 inhibitor followed by eprhinA1 (**j**–**l**). Original magnification ×200.

**Figure 3 ijms-24-03629-f003:**
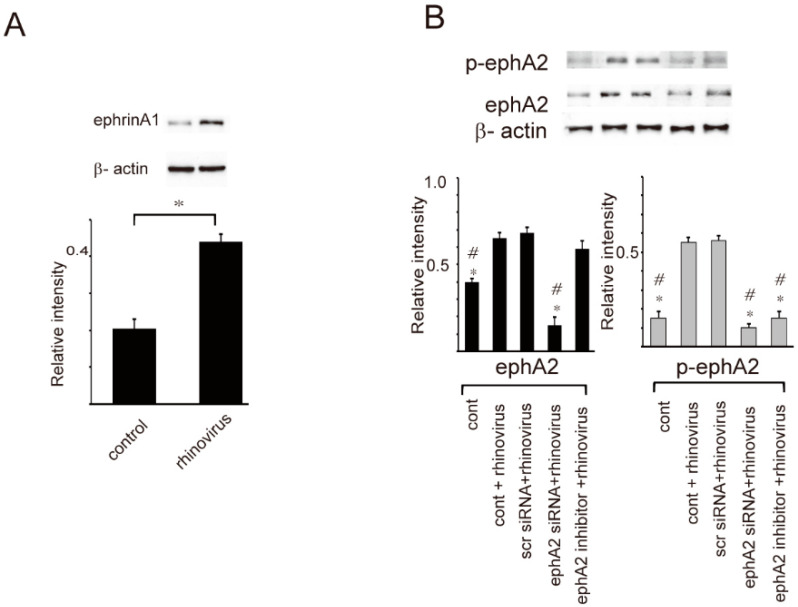
(**A**) The expression level of ephrhinA1 was increased in cultured epithelial cells treated with rhinovirus. (**B**) The expression levels of ephA2 and phosphorylated ephA2 (p-ephA2) were increased in cultured epithelial cells treated with rhinovirus (cont + rhinovirus) and in cells pretreated with scr siRNA followed by stimulation with rhinovirus (scr siRNA + rhinovirus). The levels of ephA2 and p-ephA2 were not increased in ephA2 siRNA-transfected or ephA2 inhibitor-pretreated epithelial cells followed by rhinovirus infection compared with those of cells treated with rhinovirus alone (cont + rhinovirus) or cells pretreated with scr siRNA followed by treatment with rhinovirus (scr siRNA + rhinovirus). Data from seven independent experiments are presented as means ± standard error of the mean (SEM). Upper panels show the representative western bands of ephA2 and p-ephA2. * and # in (**B**) indicate statistical significance *p* < 0.05 versus rhinovirus + control or rhinovirus + scrambled siRNA (scr siRNA).

**Figure 4 ijms-24-03629-f004:**
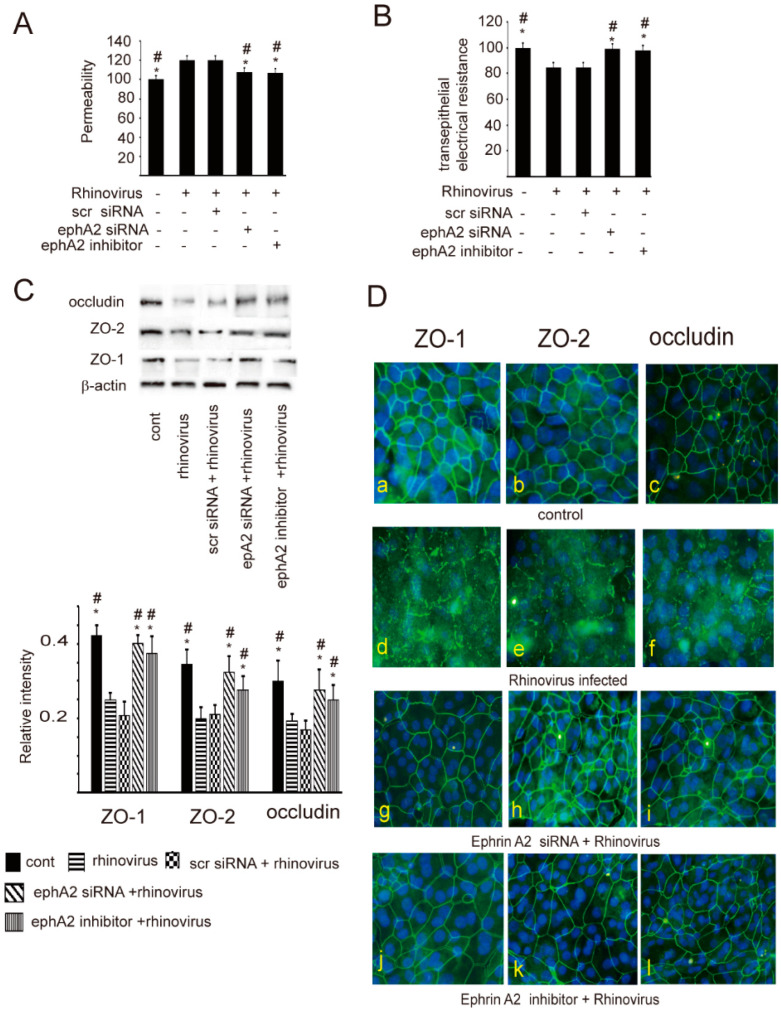
The increased epithelial permeability (**A**) associated with decreased electrical resistance (**B**) induced by rhinovirus was attenuated by blocking ephA2 with ephA2 siRNA or ephA2 inhibitor in epithelial cells. (**C**) The expression levels of ZO-1, ZO-2, and occludin were decreased by rhinovirus infection in cultured epithelial cells. Their levels were reversed by blocking ephA2 with ephA2 siRNA or ephA2 inhibitor. * and # indicate statistical significance *p* < 0.05 versus rhinovirus or scrambled siRNA (scr siRNA) + rhinovirus. The upper panels located in (**C**) show representative protein bands evaluated with western blot. All data derived from seven different subjects are presented as means ± SEM. (**D**) Confocal microscopic findings; the distribution pattern of ZO-1 (**a**,**d**,**g**,**j**), ZO-2 (**b**,**e**,**h**,**k**), and occludin (**c**,**f**,**i**,**l**) in non-treated epithelial cells (control; **a**–**c**), in cells infected with the rhinovirus (rhinovirus-infected; **d**–**f**), in cells co-treated with ephA2 siRNA transfection and rhinovirus (ephA2 siRNA + rhinovirus; **g**–**i**), or in cells co-treated with ephA2 inhibitor and rhinovirus (ephA2 inhibitor + rhinovirus; **j**–**l**). Original magnification ×200.

## Data Availability

All data are depicted in the manuscript and raw data can be additionally re requested from the corresponding author.

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
