# Peer review of "The EphA1 and EphA2 Signaling Modulates the Epithelial Permeability in Human Sinonasal Epithelial Cells and the Rhinovirus Infection Induces Epithelial Barrier Dysfunction via EphA2 Receptor Signaling"

_ijms, 2023, doi:10.3390/ijms24043629_

Round 1

Reviewer 1 Report

The introduction needs some modification with the role of Eph1 is there any balance or ratio between eph1 and 2?   What are ZO1, ZO2, and Occuludin and their significance? I think the authors should put some more details regarding this.

Some abbreviation needs to explain like TEER and ZO 1 and 2. 

Figures 1A and B need to describe the name of the treatment below the X-axis. Moreover figure 1C and D are a bit confusing and need more description of each bar and what they are showing. Figure 1 legend has some minor typographical errors (Line 67).

Figure 3 legends need some modification with more details about the experiment such as Western results etc (103-110). 

did the authors look at if you block eph1 what will be the effect on permeability?

Author Response

Re: The ephA1 and ephA2 signaling modulates the epithelial permeability in human sinonasal epithelial cells and the rhinovirus infection induces epithelial barrier dysfunction via ephA2 receptor signaling.

Dear, editor in Chief,                                           Feb. 8  2023

I thank the editors and referees of International Journal of Molecular Science for taking their time to revise my article.  I have made some corrections and clarifications in the manuscript after going over the referee’s comments.  The revised manuscript was uploaded via on line. The changes are summarized below. Revised manuscript is submitted as marked copy. In marked copy, to any text that was part of the original manuscript but has now been deleted, strike through formatting was applied. To any text that was not a part of the original manuscript but has now been added, underline formatting was applied. The font style of each comment and response was Times New Roman

Reviewer 1

1) Reviewer’s Comment

The introduction needs some modification with the role of Eph1 is there any balance or ratio between eph1 and 2?   

Authors’ response: As reviewer indicates, the role of ephA1 an dephA2 was added to the introduction which was marked as underline. (page 2; line 43-47).

Reviewer’s Comment:

What are ZO1, ZO2, and Occuludin and their significance? I think the authors should put some more details regarding this.

Authors’ response: As reviewer indicates, the detailed explanation for Zo-1, Zo-2, and occludin was added to the revised manuscript which was marked as underline. (p 8; line 239-243).

Reviewer’s Comment:

Some abbreviation needs to explain like TEER and ZO 1 and 2. 

Authors’ response: As indicates, the abbreviation for TEER, ZO-1, and ZO-2 was added to the revised manuscript which was marked as underline. (p 7; line 225, p8; line 241).

Reviewer’s Comment:

Figures 1A and B need to describe the name of the treatment below the X-axis. Moreover figure 1C and D are a bit confusing and need more description of each bar and what they are showing. Figure 1 legend has some minor typographical errors (Line 67).

Authors’ response: As reviewer indicates, the name of the treatment below the X-axis was inserted in Figure A and B. More description of each bar was added to the Figure legends of Figure 1 C and D which was marked as underline.

Reviewer’s Comment:

Figure 3 legends need some modification with more details about the experiment such as Western results etc (103-110). 

Authors’ response: As reviewer indicates, detailed explanation for the experiments were added to the Figure 3 legend which was marked as underline.

Reviewer’s comment:

did the authors look at if you block eph1 what will be the effect on permeability?

Authors’ response:

The action of ephA1 is conducted via ephA2 receptor.

In the present study, the action of ephA1 on the epithelial permeability was not evaluated with the use of ephA1 inhibitor because of the absence of ephA1 blocker. The effect of ephA1 on the epithelial permeability was elucidated by measuring the epithelial permeability in cells treated with ephA1 alone compared with those of nontreated control cells. The data was presented in Figure 1 and results section. The sentences for additional description was not added to the revised manuscript.

I look forward to hearing from you regarding the status of this manuscript.

                                             Sincerely

                   Sang Hag Lee, MD

                   Department of Otorhinolaryngology-Head & Neck Surgery

                   College of Medicine, Korea University,

                   KoreaDaeRo 73, SungBuk-Ku, Seoul, 02841, South Korea,                   

Tel: 82-2-920-5486  Fax: 82-2-925-5233

Reviewer 2 Report

This study investigated the role of ephrinA1/ephA2 signaling on sinonasal epithelial permeability and rhinovirus-induced epithelial permeability. It was concluded that the ephrinA1/ephA2 pathways could participate in the maintenance of epithelial barrier integrity in sinonasal mucosa, providing new therapeutic targets for rhinovirus-induced epithelial permeability. In the method section of permeability determination, the FITC-dextran was used. But the detailed method or the computational formula should be described in detail. Besides, only the in vitro experiments were conducted, but the in vivo experiments was not designed. If there are some experiments in vivo, the conclusion might be more convincing.

Author Response

Re: The ephA1 and ephA2 signaling modulates the epithelial permeability in human sinonasal epithelial cells and the rhinovirus infection induces epithelial barrier dysfunction via ephA2 receptor signaling.

Dear, editor in Chief,                                           Feb. 8  2023

I thank the editors and referees of International Journal of Molecular Science for taking their time to revise my article.  I have made some corrections and clarifications in the manuscript after going over the referee’s comments.  The revised manuscript was uploaded via on line. The changes are summarized below. Revised manuscript is submitted as marked copy. In marked copy, to any text that was part of the original manuscript but has now been deleted, strike through formatting was applied. To any text that was not a part of the original manuscript but has now been added, underline formatting was applied. The font style of each comment and response was Times New Roman

Reviewer 2

This study investigated the role of ephrinA1/ephA2 signaling on sinonasal epithelial permeability and rhinovirus-induced epithelial permeability. It was concluded that the ephrinA1/ephA2 pathways could participate in the maintenance of epithelial barrier integrity in sinonasal mucosa, providing new therapeutic targets for rhinovirus-induced epithelial permeability.

Reviewer’s comment:

In the method section of permeability determination, the FITC-dextran was used. But the detailed method or the computational formula should be described in detail.

Author’s response: As reviewer indicates, the detailed method was added to the revised manuscript which was marked as underline. (p 7, line 231-236).

Reviewer’s comment:

Besides, only the in vitro experiments were conducted, but the in vivo experiments was not designed. If there are some experiments in vivo, the conclusion might be more convincing.

Author’s response:

As reviewer indicates, some limitations exist in this study. The effect of ephrinA1 and ephA2 receptor pathways on the epithelial permeability needs to be confirmed in in vivo experiments, including animal studies. These sentences were added to the conclusion section of revised manuscript which were marked as undeline. (p 8; line 279-281).

I look forward to hearing from you regarding the status of this manuscript.

                                             Sincerely

                   Sang Hag Lee, MD

                   Department of Otorhinolaryngology-Head & Neck Surgery

                   College of Medicine, Korea University,

                   KoreaDaeRo 73, SungBuk-Ku, Seoul, 02841, South Korea,                   

Tel: 82-2-920-5486  Fax: 82-2-925-5233
